# Learning and Applying Cooperative Solutions: A Classroom Experiment on Transportation Games

**Nikolaos Georgantzis [1,2,*], Carlos Gutiérrez-Hita [3] and Joaquín Sánchez-Soriano [4]**

1   School of Wine and Spirits Business, Burgundy School of Business, 21000 Dijon, France
2   Laboratorio de Economia Experimental, Jaume I University, 12076 Castellón de la Plana, Spain
3   Center of Operations Research, University Miguel Hernández, Av/Universidad, s/n, Edificio Torretamarit, 03202 Elche, Spain
4   Department of Statistics, Mathematics and Computer Science, Avinguda de la Universitat d'Elx, 03202 Elche, Spain
*   Correspondence: nikolaos.georgantzis@bsb-education.com

**Abstract:** In a trade experiment, groups of students were taught how to bargain over a pie generated in a transportation game. Data collection and detailed group reports of the bargaining process allowed us to identify the type of bargaining followed and its correspondence with cooperative game theory concepts. Explicit coalitions were rather scarce (17%), although coalition stability was implied by 47.8% of the agreements. Efficiency was achieved in the vast majority (82%) of cases, whereas in 34.8% of the agreements, students used a lexicographic ordering of multiple solutions before choosing among them. Regarding the bargaining procedure, in 40% of the agreements, quantities traded were decided before profit sharing rules were negotiated, whereas in 16% of the cases they were simultaneously agreed upon. Our findings suggest that bargaining procedures often do not imply explicit coalitions. Moreover, efficiency can be achieved even in the absence of bargaining processes.

**Keywords:** learning; classroom experiment; bargaining; transportation games

## 1. Introduction

The non cooperative human behavior in bargaining contexts has been extensively studied in the framework of simple settings such as the ultimatum game (a vast experimental literature on ultimatum games was inspired by [1]). Nevertheless, in most cases, bargaining in real situations often involves long and complex procedures. Furthermore, together with the splitting of the pie among parties, bargaining also affects the overall efficiency of a trading game. We propose a transportation game to study the extent to which experimental subjects use cooperative solutions in bargaining contexts. Indeed, although there are other approaches to learning bargaining rules, transportation problems are useful, as they emerge in several economic transactions. For instance, retailers usually sign contracts with wholesalers that involve not only a quantity of input but also the arrangements concerning the transportation procedure and the extra cost that it involves. In a similar way, international trade contracts involve bargaining procedures that include agreements that specify timing, shipping cost, transportation insurance and the quantity traded.

Within this strand of the literature, in Hong [2], students played a classroom trading game where each group was assigned to a country. Students learn the mutual benefits of trade and those factors that affect the distribution of the pie. In Swope et al. [3], agents were asked to agree on a credible bargaining position in a laboratory bargaining experiment. They allowed for both *sincere* and *strategic* respondents. In this setting it was found that, contrary to standard game theory predictions, commitment could be used effectively to

increase agent's payoff. It was also found that commitment may increase the number of failed agreements and reduce the overall efficiency from exchange. Cooperative game theoretic concepts have already been applied to the particular framework of transportation situations (for instance, in a transport game framework, cost allocation issues were studied in Samet et al. [4], whereas in Sánchez-Soriano et al. [5], core concepts were revised). In Sánchez-Soriano [6,7], the so-called pairwise solutions and their relations with the core of the associated transportation games are established. In such settings, these outcomes are based on the optimal solutions of the corresponding transportation problem (see, for instance, Thompson [8] and Sánchez-Soriano and Fragnelli [9]). In López-Paredes et al. [10] it is also pointed out the relevance of game theory in providing conceptual support to analyze strategic decisions and bargaining. Moreover, they stressed that the social dimension of the bargaining process is important in the determination of an equilibrium. In this respect, they remarked on the importance of group rationality and the learning concept in the bargaining process. Regarding our approach, few experiments have also been run on transportation problems. For instance, in Gaker et al. [11], the role of better understanding players' bounded rational behavior was studied in transport planning. Their results can serve to increase our ability to predict and influence behavior.

In our context of a trading experiment, we propose transportation games as a paradigm for the study of the learning process. Our experiment can also be seen as multi-player transportation situation in the classic operations research sense. In our experiment, the number of players within each group was arbitrarily varied in order to create a complete and well-defined spectrum of cooperative decision-making domains; the process adopted and the results reached can be meaningfully mapped (for a survey on descriptive approaches to cooperation see Selten [12]. Students were endowed with the following information: production available, marginal revenue, consumers' demand and the theoretical optimal solutions. We also ask students to deliver a written report, including relevant information about the procedures they followed to achieve agreements and the split of profits. Together with the data collected, we also compiled information about the bargaining processes and the variety of game concepts involved in them. Hence, interactive qualitative and quantitative analyses are used to analyze the behavior followed by a student–subject pool, focusing on the extent to which they want to cooperate with each other in order to maximize profits. We identify equilibrium terms of the transportation game where both fairness and efficiency concerns were relevant.

The rest of the paper is organized as follows. Section 2 provides some preliminaries on transportation games. Section 3 describes the setup of the experiment. In Section 4, we analyze the main results. Section 5 discusses some useful economic insights. Finally, Section 6 concludes.

## 2. Preliminaries: Transportation Games

Transportation games are characterized by a set of particular features. In what follows, we present useful concepts that provide the theoretical benchmark of our experiment.

### 2.1. Basic Concepts

Cooperative game theory models situations of conflicts of interest in which agents can benefit by collaborating and binding agreements are possible. In this subsection, we present some basic cooperative game theory concepts which will be used throughout the paper.

#### 2.1.1. Transferable Utility Game (Tu-Game)

It is a pair $(N, v)$, where $N = \{1, 2, \ldots, n\}$ is the set of players and $v$ is a function that represents the gain that a given coalition $S$ can obtain by itself, $S \subset N$. Given $(N, v)$, a distribution among the players is a vector $x \in \mathbb{R}^N$ such that $\sum_{i \in N} x_i \leq v(N)$. We denote by $G^N$ the set of all TU-games with set of players $N$.

### 2.1.2. Imputations of the Game

A distribution $x$ is an imputation of the game $(N, v)$ if it satisfies two conditions:

- [Individual rationality]: $x_i \geq v(\{i\})$, $i = 1, 2, \dots, N$.
- [Efficiency]: $\sum_{i \in N} x_i = v(N)$.

### 2.1.3. Core of the Game

When joining the coalition yields at least as much as the corresponding stand-alone surplus, $x(S) \geq v(S)$, then we obtain the core of the game (Gillies [13]). $x(S)$ is a distribution among players given a coalition $S \subset N$. The core is the set of all imputations that satisfy coalition rationality, or in other words the set of all coalition stable distributions satisfying efficiency, which is defined as:

$$Core(v) = \{x \in \mathbb{R}^N : x(S) \geq v(S), \forall S \subset N, \ x(N) = v(N)\}$$

Notice that if a distribution is coalition-rational, then it is individually rational.

### 2.1.4. Equity Principle

In our setting, it is natural to obtain distributions between players, a concept deeply studied in early economic literature (see for instance Homans [14]). This is applicable to situations where a bundle has to be distributed: a player $i$ receives a share $x_i$. The shares are determined according to a standard of distribution, which is a function $(r_1, \dots, r_n)$, which measures the reward for each possible distribution that each player receives, and a standard of comparison which determines the weight of the players $(w_1, \dots, w_n)$. This terminology was introduced by Selten [15]. The equity principle proposes a distribution such that the shares are proportional to the weights:

$$\frac{r_1}{w_1} = \frac{r_2}{w_2} =, \dots, = \frac{r_n}{w_n}.$$

### 2.1.5. Marginal Contributions

Given a coalition $S \subset N$, and $i$ is not included in $S$, the marginal contribution of player $i$ to coalition $S$ is given by $v(S \cup i) - v(S)$; i.e., the marginal contribution of a player to a coalition is defined as the surplus that a coalition gains when a player joins it. This concept is very relevant in game theory and was introduced by Shapley to define the well-known Shapley value of a game (Shapley [16]).

### 2.1.6. Lexicographic Orders

Given $(x, y) \subset 211d^N$, we say that $x$ is lexicographically better than or equal to $y$ ($x \succeq lex\ y$), iff $x = y$ or there is $1 \leq t \leq n$ such that $x_i = y_i$ for all $i < t$ and $x_t > y_t$. On the other hand, given a distribution $x$ of a game $(N, v)$, we define the excess of coalition $S$ at $x$ as $e(S, x) = v(S) - x(S)$,, i.e., the dissatisfaction of the members of $S$ from the distribution $x$. Here, $e(x)$ denotes the sequences of excesses of all coalitions at $x$ in decreasing order. Thus, we can say that $x$ is better than $y$ if $e(y) \geq lexe(x)$; i.e., the dissatisfaction at $y$ is higher than the dissatisfaction at $x$. These concepts are used to define the nucleolus of the game (Schmeidler [17]).

### 2.2. An Example: Cartel Problem

Suppose that three firms producing homogeneous good behavior as a cartel. They agree to reduce the total production to 600 units and decide firms' output quotas. If the standard of distribution is the output quota of each firm $i$ and the standard of comparison is the capacity of production ($C_i$), the resulting quotas by application of the equity principle are shown in Table 1. If the standard of distribution is $r_i = x_i - C_i$, the equity principle provides the same result. However, if the standard of comparison is had with *all weights being equal* , firms' quotas obtained by applying the equity principle are shown in Table 1.

Therefore, it is observed that different combinations of standards of distribution and comparison determine different assignments when the equity principle is implemented.

**Table 1.** Cartel Problem.

| Cartel Problem | | | |
|---|---|---|---|
| *Firm* | *Capacity $C_i$* | *Quota* | |
| I | 500 | 300 | |
| II | 300 | 180 | |
| III | 200 | 120 | |
| Cartel problem, II | | | |
| *Firm* | *Capacity* | $r_i = x_i$ | $r_i = x_i - C_i$ |
| I | 500 | 200 | 1100/3 |
| II | 300 | 200 | 500/3 |
| III | 200 | 200 | 200/3 |

## 3. The Experiment

### 3.1. Set Up

In our experiment, students were split into two groups. Students in the set $P$ were called producers (sellers), whereas those in the set $R$ were called retailers (buyers). Each producer $P_i \in P$ has a non negative number of units of a certain homogeneous and indivisible good $(p_i)$, and each retailer (buyer) $R_j \in R$ demands at most a non negative number of units of that good $(q_j)$. Each seller and each buyer may conduct several trade agreements $x_{ij} = \min = \{p_i, q_j\} \geq 0$. The trading of one unit between a producer–retailer pair $P_i R_j$ produces a total non-negative profit $b_{ij} \geq 0$. Let $B$ be the $n \times m$ matrix of profits, where $p$ is the $n$-dimensional supply vector of units available from the producers (sellers) and $q$ is the $m$-dimensional vector of demands.

**Definition 1.** *In our context, a transportation situation is defined by $(P, R, B, p, q)$.*

Players have to divide the pie using transfers of profits (utility) within producer–retailer pairs who are matched in a feasible (optimal) solution. Sharing procedures are used to generate a set of distributions (units of output traded) among the bargaining groups for each feasible (optimal) solution. Finally, a distribution of profits is obtained through bargaining between all agents involved. When the bargaining process and any profit transfers take place within each producer–retailer pair, the distribution of the total profit obtained is called a pairwise solution (Sánchez-soriano [6,7]).

We identify every transportation situation $(P, R, B, p, q)$ with its associated transportation game, a TU-game $v \in G^N$. For every coalition $S \subset N$ with producers (sellers) $S_P = S \cap P$ and retailers (buyers) $S_R = S \cap R$, we define the transportation game as follows:

- The set of players is $P \cup R$.
- The characteristic function is given for every $S \subset N$ by

$$
v(S) = \begin{aligned} & \max_{x_{ij}} \sum_{i \in S_P} \sum_{j \in S_R} b_{ij} x_{ij} \\ s.t \quad & \begin{cases} \sum_{j \in S_R} x_{ij} \leqslant p_i, \ i \in S_P \\ \sum_{i \in S_P} x_{ij} \leqslant q_j, \ j \in S_R \\ x_{ij} \geq 0, \ (i,j) \in S_P \times S_R \end{cases} \end{aligned} \tag{1}
$$

When $S \subset P$ or $S \subset R$, the value of the characteristic function is $v(S) = 0$. We identify $(P, R, B, p, q)$. We denote by $TG^{P \cup R}$ the class of transportation games between a set of students $P \cup R$ and by $TG$ the class of all transportation games.

### 3.2. Participants

The experiment was carried out with 113 students at the University of Alicante. Sixty-five students were enrolled in the third year of *Diplomatura en Graduado Social*, and 48 students were enrolled in the second year of *Diplomatura de Relaciones Laborales*. They are similar

three-year degrees on labor law and labor relations and human resource management. In our transportation context, we want to observe the extent to which students learn to bargain in situations involving a conflict of interest. Students' rewards came from their experimental profits and also from the quality of the surveys they reported. Participants lived in an urban environment with similar cultural backgrounds. Participants were divided into 25 groups. The size of the groups was intended to facilitate interaction among players and to avoid the possible complexity derived from an excessively high number of experimental subjects. The distribution was as follows: 5 groups with 3 players $[1, 2]$, 9 groups with 4 players (7 groups as $[2, 2]$ and 2 groups as $[1, 3]$ and $[3, 1]$), 7 groups with 5 players (5 groups as $[3, 2]$ and 2 groups as $[2, 3]$), 2 groups with 6 players $[3, 3]$, one group with 7 players $[3, 4]$ and one group with 8 players $[4, 4]$.

### 3.3. Procedures

Detailed instructions for students are reported in the Appendix A. We include here the general procedure of the experiment:

1. Relevant elements of a transportation situation were explained in the classroom: the role of each student (producer or retailer), how profits are generated, production and demand vectors, and the nature of the output traded (indivisible and homogeneous). The feasibility and optimality of solutions in a transportation situation were also explained.
2. Students were allocated to groups with the following information: (i) producer's capacity; (ii) retailer's demand; and (iii) the matrix of unitary net profits—i.e., the highest net profit they could obtain when a unit of output was traded (see Table A and Picture A in the Appendix A.
3. Each player's goal was to obtain the maximum profit. They could sign binding collaboration agreements with other players, they could cooperate or compete among each other and they could follow any other procedure or mechanism (cooperative or competitive) in order to obtain their goals. Indeed, there was not any practical restriction in the behavior of the players except feasibility, $\sum_{i \in N} x_i \leq v(N)$.
4. The experiment was extended over 9 weeks. At the end, each group had to deliver a written survey including: (i) a detailed description of the collective and/or individual procedures they followed to achieve the final result; (ii) final trading, if any; and (iii) final profit sharing rule.
5. Students' payoffs. Up to 2 points (out of 10) of the final mark in the subject *statistics* If the students did not achieve any agreement they obtained zero points in the profit-contingent part of the aforementioned reward. The quality of the report submitted accounted for 75% of the total reward, whereas individual profit obtained as a result of the allocation and negotiation procedures was 25%.

### 3.4. Data Collection and Qualitative Variables

The data reported here are the results of the quantitative outcomes and the written survey delivered by the students according to the instructions. Apart from the quantitative outcomes, we were interested in the qualitative variables presented in Figure 1 in order to look for evidence on game theory concepts followed by the students in the learning process. Next, we explain the meaning of each variable in the written surveys delivered by the students.

| Group | Variable | Categories |
|---|---|---|
| **Behavioral** | Behavior (BEH) | Cooperate/Compete/Negotiate/Others |
| **Game theoretical ideas arising** | Individual rationality (IR) | Yes/Similar/No |
| | Coalitional rationality (CR) | Yes/Similar/No |
| | Marginal contributions (MC) | Yes/Similar/No |
| | Lexicographical orders (LEX) | Yes/Similar/No |
| | Coalition formation (CF) | Yes/No |
| **Control** | SIZE | Up to 5 players/More than 5 players |
| | Symmetry (SYM) | Both sides/One side/No symmetry |
| | Multiple optimal solutions (MS) | Yes/No |
| **Cooperative results** | Efficiency (EFF) | Yes/No |
| | Coalitional stability (ST) | Yes/No |
| **What did they use?** | Optimality (OPT) | Yes/No |
| | Pairwise bargaining (PS) | Yes, but egalitarian/Yes, but no egalitarian/No |

**Figure 1.** Categories considered for each variable.

In the first column of , we show the five groups of variables we have considered. The first group, behavioral, only has one variable, behavior. This variable shows what kind of behavior was used by the students when facing the problem of distributing the pie among them. The second group, game theoretical ideas arising, consists of five variables, one for each of five cooperative game theory concepts. In this case, all variables have three categories, except one that is dichotomous. These variables indicate whether these concepts were used in some way. The third group of variables is control, which is made up of variables whose values do not depend on what the students do, but on the design of the problem that has been proposed to them. The fourth group of variables evaluates if the distribution obtained has any property related to properties of distributions in cooperative games. In particular, two variables related to the concepts of coalition efficiency and stability are observed. The last group of variables has to do with what the students used to reach a final distribution of the benefit generated. In this case, two variables were observed that measured whether they had used optimal solutions and pairwise bargaining to achieve a benefit distribution. Each variable category is explained in more detail below.

First, we focus on students' behavior. We considered that (i) students cooperated if they looked jointly for a distribution of the total profit according to any criteria; (ii) students competed if they reacted as a result of the strategies that the other players followed; (iii) students negotiated with each other in order to agree on the final allocation of the total profit generated. The cases in which students did not follow any criteria were categorized as *other*.

Second, we checked the extent to which specific game-theoretic concepts arose in the bargaining processes followed by students. Notice that although we could include efficiency here also, it is included in another set of variables.

In particular, we were interested in *individual rationality* (whether students analyzed how much was the least they could guarantee themselves); *coalition rationality* (whether coalitions analyzed how much was the least they could guarantee themselves); *marginal contributions* (whether students analyzed their contributions to a coalition or something similar); *lexicographical orders* (we observed whether students applied some kind or order

criterion in the selection of a distribution when there were several alternatives); and *coalition formation* (whether students coordinated with binding agreements).

Third, as control variables we considered the *size* (number of players by group), *symmetry* (two-sided symmetry—both producers and retailers are symmetric; one-sided symmetry—producers or retailers were symmetric alone; and heterogeneity on both sides of the market) and *multiplicity of optimal solutions* (unique or multiple optimal solutions). In our context, symmetry means that an agent has the same units for sale (producer) or demand (retailer).

Fourth, we considered two concepts which are closely related to cooperative game theory: efficiency and coalition stability (i.e., the core of the game). We were interested in whether: (i) the final distribution is efficient and (ii) whether it belongs to the core of the game. In our context, *efficiency* means that a group of students reached an aggregate profit larger or equal than the profit obtained in any other coalition (subgroup) of students. *Coalition stability* exists when the final distribution of the total profit is in the core of the corresponding game.

Finally, we were interested in the way students would bargain over the distribution of total profit. In our context we would observe *optimality* if students implemented an optimal solution—i.e., when they agreed about the distribution of the overall profit. *Pairwise bargaining* would exist if students only used bargaining within producer–retailer pairs, in the absence of side payments (apart from producer–retailer trading). Moreover, in the case of pairwise bargaining, we distinguished whether the distribution of the profit was egalitarian or not.

## 4. Results

We first present a general picture of the main insights. Second, a detailed discussion of selected cases is provided (see Tables 2 and 3).

### 4.1. General Results

Out of the 25 groups initially formed, 23 finally reached an agreement and successfully submitted a survey. Table 2 summarizes the results. The most frequent process was a two-stage negotiation. It was followed by 10 groups. In a first stage, the pattern of trade was established, reaching in the majority of cases an optimal solution. In the second stage, negotiations took place within producer–retailer pairs to determine the split of the surplus generated from trade. The second and third most frequent patterns followed were the simultaneous determination of quantities traded and surplus sharing (four groups), or directly, a negotiation on the split of the surplus corresponding to the optimal solution (four groups). In fewer cases (three groups), coalitions were formed to decide collectively on the quantities and shares for a single player. Finally, two groups decided cooperatively on the quantity traded, although negotiation took place individually within producer–retailer pairs.

**Table 2.** Bargaining procedures adopted by players.

| Bargaining Procedure | # of Groups |
|---|---|
| 1st quantities-2nd sharing by pairs | 10 |
| Simultaneous quantities and sharing by pairs | 4 |
| Optimal solution and then bargaining by pairs | 4 |
| Collective bargaining on surplus sharing | 3 |
| Cooperation, but surplus sharing by pairs | 2 |
| Analysis of alternatives-No agreement | 2 |

Relevant cooperative game-theoretic concepts, such as individual rationality, coalition rationality and marginal contributions, were totally absent from the reports submitted. Only in eight reports bargaining followed a lexicographic ordering of solutions, whereas in

four cases, coalitions were explicitly mentioned. In 19 reports the outcome reached was an efficient solution. Moreover, in 11 groups, the split of profits satisfied the core of the underlying game. Table 3 summarizes.

**Table 3.** Game-theoretic ideas implied.

| Concept | %Success |
| --- | --- |
| Coalition stability | 47.8% |
| Efficiency | 82.6% |
| Coalition formation | 17.4% |
| Lexicographical orders | 34.8% |
| Marginal contributions | 0% |
| Coalition rationality | 0% |
| Individual rationality | 0% |

Regarding the emergence of cooperative results, in 19 groups (82.6% of the cases) the final distribution of surplus was efficient. As we will argue below, the four cases where the distribution of surplus was inefficient relate to the nature of the transportation problem. Despite the fact that no reference was made in any of the reports to any concept analogous to coalition rationality, in 11 groups (47.8%), distributions were in the core of the corresponding cooperative game. It is also interesting to note the pervasive use of pairwise negotiations, observed in 20 groups (86.9%).

*4.2. Analysis of Relations between Variables*

In this subsection we explore the extent to which variables defined in the experiment are related with each other. To do so, we follow two measures. First, we present the Lambda Goodman and Kruskal analysis (Goodman and Kruskal [18]), which reports the predictive capacity between variables. Second, we present the values of Cramèr's V ([19]), reporting the association (if any) between variables. Both techniques are presented in the symmetric and the asymmetric versions. The asymmetric version ) matches each explanatory variable (our control variables) and each response variable (the rest of the variables). In the symmetric version, cross-relations between response variables were measured.

Lambda values (see Tables 4 and 5) account for the reduction in the prediction error when there is perfect knowledge of other variables involved in the study. It takes values on the unit interval, where *zero* represents a null reduction and *one* a full reduction. The values of Cramèr's V (see Tables 6 and 7) also lie between *zero* (which means independence) and *one* (which means full association). In brackets, the *p*-values corresponding to the Chi-square tests. We also parametrized values inside the interval (according to the values obtained): values in $(0, 0.4]$ imply a low association; values between $(0.4, 0.6]$ an intermediate or moderate association; values in $(0.6, 0.8]$ imply a high association; finally values, in $(0.8, 1]$ imply a very high association.

Tables 4 and 6 contain the asymmetric analysis (i.e, the relation between control and other variables). First, it is interesting to note that when the transportation problem has multiple solutions, we can reduce the prediction error when pairwise bargaining takes place by 25% (see Table 8). Therefore, multiple solutions and pairwise bargaining are somehow related. This is also confirmed when we look at Table 6, where Cramèr's V (0.50) reveals a statistically significant association between these two variables. Second, the relation between MS and LEX is also relevant. Although the Goodman and Kruskal's asymmetric Lambda is zero (which means that ex ante knowledge of the MS category does not reduce the prediction error of the most likely result in variable LEX), Cramèr's V (0.44) reveals a moderate statistically significant association between these two variables. The intuition behind this is that when there are multiple optimal solutions, agents try to sort them in order to reach an agreement.

**Table 4.** Goodman and Kruskal's asymmetric lambda.

| $\lambda_a(\%)$ | EFF | ST | PS | OPT | CF | LEX |
|---|---|---|---|---|---|---|
| SIZE | 0.25 | 0.09 | 0.08 | 0.14 | 0 | 0 |
| MS | 0 | 0.27 | 0.25 | 0 | 0 | 0 |
| SYM | 0 | 0.27 | 0 | 0 | 0 | 0 |

**Table 5.** Goodman and Kruskal's symmetric lambda.

| $\lambda(\%)$ | EFF | ST | PS | OPT | CF | LEX |
|---|---|---|---|---|---|---|
| EFF | * | | | | | |
| ST | 0.20 | * | | | | |
| PS | 0.06 | 0.04 | * | | | |
| OPT | 0.45 | 0.22 | 0.16 | * | | |
| CF | 0 | 0.07 | 0.38 | 0.18 | * | |
| LEX | 0.08 | 0.16 | 0.15 | 0.07 | 0 | * |

**Table 6.** Cramèr's V (Asymmetric).

| $V$ | EFF | ST | PS | OPT | CF | LEX |
|---|---|---|---|---|---|---|
| SIZE | 0.52 (0.098) | 0.17 (0.889) | 0.36 (0.418) | 0.38 (0.347) | 0.34 (0.459) | 0.35 (0.469) |
| MS | 0.10 (0.624) | 0.30 (0.147) | 0.50 (0.056) | 0.14 (0.493) | 0.37 (0.078) | 0.44 (0.104) |
| SYM | 0.01 (0.957) | 0.40 (0.076) | 0.40 (0.164) | 0.25 (0.226) | 0.25 (0.231) | 0.09 (0.914) |

**Table 7.** Cramèr's V (symmetric).

| $V$ | EFF | ST | PS | OPT | CF | LEX |
|---|---|---|---|---|---|---|
| EFF | * | | | | | |
| ST | 0.44 (0.035) | * | | | | |
| PS | 0.26 (0.461) | 0.15 (0.781) | * | | | |
| OPT | 0.69 (0.061) | 0.44 (0.038) | 0.61 (0.014) | * | | |
| CF | 0.21 (0.313) | 0.21 (0.315) | 0.85 (0.000) | 0.44 (0.033) | * | |
| LEX | 0.52 (0.043) | 0.34 (0.260) | 0.35 (0.241) | 0.42 (0.128) | 0.18 (0.700) | * |

**Table 8.** Game-theoretic ideas implied.

| Concept | %Success |
|---|---|
| Coalition stability | 47.8% |
| Efficiency | 82.6% |
| Coalition formation | 17.4% |
| Lexicographical orders | 34.8% |
| Marginal contributions | 0% |
| Coalition rationality | 0% |
| Individual rationality | 0% |

Tables 4 and 5 include the descriptive analysis of the remaining variables. We can observe some ex ante predictable relations between variables EFF and OPT; EFF and ST; and OPT and ST. In particular, the intensity between CF and PS is somehow spurious, due to the fact that, when a coalition is formed, the split of profit is not negotiated by pairs, whereas when a coalition is not achieved we observe a negotiation by pairs. More interesting relations are observed between PS and LEX. This can be explained because players who used some lexicographic idea also were involved in a negotiation of profits within producer–retailer pairs. Finally, we point out the relation between OPT and PS. This fact arose because an optimal solution always yields a pairwise negotiation.

### 4.3. Students' Behavior

We report here students' behavior. In our context, behavior describes the types of strategies that a student (or a coalition of students) followed when they defined a trading plan in order to capture as much as profits they could. We focus on the most interesting markets to highlight two facts: (1) how students manage conflict situations; and (2) it was not an objective of students to achieve a stable result. From an economic perspective, the indication is that students were not rational (i.e., they were not strictly profit maximizing). They used rather weaker principles instead of rationality, such as equity, reasonableness and accessibility. Moreover, players did not provide a stable result in general (notice that students in two markets did not achieve any result of distribution of profit after several rounds of negotiations. Thus, we are not able to evaluate any idea related to coalition rationality in those markets).

**Market 5**

|      | *R1* | *R2* |    |
|------|------|------|----|
| *P1* | 7    | 8    | 10 |
|      | 7    | 6    |    |

**Market 8**

|      | *R1* | *R2* | *R3* |    |
|------|------|------|------|----|
| *P1* | 2    | 2    | 2    | 10 |
| *P2* | 2    | 2    | 2    | 16 |
|      | 8    | 4    | 4    |    |

**Market 15**

|      | *R1* | *R2* |    |
|------|------|------|----|
| *P1* | 2    | 2    | 10 |
| *P2* | 2    | 2    | 8  |
| *P3* | 2    | 2    | 12 |
|      | 14   | 14   |    |

**Market 16**

|      | *R1* | *R2* | *R3* |    |
|------|------|------|------|----|
| *P1* | 2    | 2    | 2    | 10 |
| *P2* | 2    | 2    | 2    | 16 |
|      | 8    | 4    | 4    |    |

**Market 18**

|      | *R1* | *R2* |   |
|------|------|------|---|
| *P1* | 7    | 6    | 2 |
| *P2* | 5    | 4    | 3 |
| *P3* | 6    | 5    | 4 |
|      | 3    | 5    |   |

Students in Market 5 started by considering the four possible combinations of trading with an egalitarian distribution of profits:

| $(x_{11}, x_{12})$ | $(7, 3)$ | $(6, 4)$ | $(5, 5)$ | $(4, 6)$ |
|-------------------|-----------|-----------|-------------|-----------|
| Distribution | (36.5;24.5,12) | (37;21,16) | (37.5;17.5,20) | (38;14,24) |

Secondly, they alternatively began to submit offers and counteroffers consisting on a trading plan and a profit distribution between producer–retailer pairs (this procedure was also used in three other markets). Only trading plans $(7, 3)$ and $(4, 6)$ were considered. After 23 bargaining rounds, they agreed to implement trading plan 4 with a distribution of profits $14 + 14$ for $P1$ and $33 + 15$ for $R1$ and $R2$. However, $R1$ and $R2$ disagreed with the result; therefore, $P1$ offered to retailers half of their losses with respect to their best options. Finally, the distribution of profits was 8.75 and $(14 + 5.25)$ for $P1R1$; and for $P1R2$ was 28.5 and $(15 + 4.5)$; hence, there was a pie of $(37.25; 19.25, 19.5)$. This agreement corresponds to the equity principle in the following way:

| *Player* | $\omega_i = M_i$ | $r_i = M_i - x_i$ | $r_i / \omega_i$ |
|----------|------------------|-------------------|------------------|
| *P1*     | 47               | 9.75              | 0.21             |
| *R1*     | 24.5             | 5.25              | 0.21             |
| *R2*     | 24               | 4.5               | 0.19             |

where the standard of comparison is the best result obtained by each player throughout the negotiation process. Likewise, we observed that in the last step of the negotiation process, there were transference of profits (utility) from $P1$ to $R1$ and $R2$. However, such a transfers were carried out through producer–retailer pairs.

Market 8 has only one optimal trading plan. Students took as a given this single plan, because it was the only way to obtain the maximum total profit. Hence, they negotiated between each producer–retailer pair under an optimal trading plan. This procedure was also used in another three markets. In this particular case, the final split of profit was the egalitarian distribution between each producer–retailer pair $(6.0, 8.5; 6.5, 8.0)$.

Students in Market 15 agreed to cooperate and to distribute the total profit. As the total production was two units higher than the total demand, they implemented the trading plan symmetrically on both sides: $PiRj = 5$, $P2Rj = 4$, $(i = 1, 3)$ and $(j = 1, 2)$. However, $P3$ lost out with it. Hence, they decided to divide the profit generated in each producer–retailer pair in an egalitarian way, except between pairs $P3R1$ and $P3R2$; in those cases, $P3$ was endowed with 5.5, and $Rj$ was endowed with 4.5. Thus, the final distribution was $(10, 8, 11; 13.5, 13.5)$. This was done in order to compensate $P3$, and thus, students explicitly considered transference of utility (money). Something similar happened in another market with the same structure. However, in that case, students implemented the following trading plan in order to avoid complaints by the players: $PiRj = 5$, $(i = 1, 2, 3)$ and $(j = 1, 2)$. The profit generated was 48. It was distributed in an egalitarian fashion. In order to maintain the structure of the problem, they divided the profit generated by each producer–retailer pair into $4.8 + 3.2$. Once again, the players explicitly considered the possibility of transfer of utility (money) but respecting the matches given by the implemented trading plan.

In Market 16 students bargained for only one round. Then, retailers formed a coalition with a leader in order to negotiate with the producers. Finally, the retailers' leader achieved a separate agreement with each producer, and the total profit obtained was split proportionally to their demands within the coalition. The final distribution of the total profit was $(8.1, 5.58; 9.16, 4.58, 4.58)$. Something similar happened in another two markets (in both cases with one producer and two retailers), but the total profit was first equally divided between the producer and the retailers, and thus split in the coalition proportionally to each demand.

Finally, the scheme in Market 18 had eight possible optimal trading plans. First, each student considered the eight optimal trading plans in terms of individual preferences according to their profit contributions in each optimal trading plan (the larger the contribution, the higher the opportunity of business). Thus, they obtained a system of preferences linked with the optimal solutions: the optimal solution minimizing the maximal loss (complaint). Therefore, they applied a lexicographic criterion to determine the implemented trading plan (something similar was also used in another 7 markets). Once the trading plan (in this case an optimal solution of the problem) was selected, they proceeded to negotiate the distribution of the profit by matching producer–retailer pairs in the selected trading. The following distribution of the total profit was obtained: $(6.430, 4.445, 10.625; 9.000, 12.500)$. This procedure, selecting first the trading plan and then bargaining within each producer–retailer pair, was also used, in different ways, in another nine markets.

## 5. Discussion of the Results

In this section, we want to point out some observations based on the results highlighted in previous section. Overall, we found that in the majority of cases (83%), experimental subjects used a sequential bargaining procedure (quantities first, and then surplus sharing). Moreover, the egalitarian solution involving a pairwise egalitarian distribution emerged in 39% of cases. Only in 17% of cases did we observe the explicit formation of coalitions among players, no matter the number of players in the market. Finally, it is interesting to note the use of a quasi-lexicographic order to select a distribution of the overall profit, particularly when there are multiple optimal solutions.

Moreover, we found that players' behavior was conditioned by the structure of the problem. In fact, up to 82.6% of the markets reaching an agreement (two markets did not reach any) used optimal and/or feasible solutions as a starting point before splitting the bundle generated. Hence, the way in which players solved the associated optimiza-

tion problem was relevant for them, also indicating that they understood the underlying economic problem.

Furthermore, up to 86.9% of the markets that reached a split of the bundle generated took place under negotiations between producers and retailers. In most cases, negotiations accounted for the associated optimal and/or feasible solutions. It is also relevant that side payments (utility transfers) did not occur outside the producer–retailer pairs. This result is in line with previous literature (see, for instance, Shapley and Shubik [20], Thompson [8], and [6,7]).

We have identified a number of negotiation procedures repeated across the experiments. Among them, three procedures emerged as the most frequently used: (i) first–second quantity sharing by pairs (40%); (ii) simultaneous quantities and sharing by pairs (15%); and (iii) optimal solution and then bargaining by pairs (15%). In the three categories, negotiations under producer–retailer pairs were observed.

Finally, it is remarkable that, in general, basic concepts of classical cooperative game theory did not appear in the learning procedures observed along the rounds of the experiment. The most remarkable exceptions were the concept of *coalition formation* that was observed in 4 markets, and lexicographic orders. The later was not expected ex ante, due to the more complicated economic procedures implied.

## 6. Conclusions, Recommendations and Limitations

In this paper, we analyzed the results obtained in a series of classroom experiments where groups of students traded in the context of a transportation problem. Cooperative game theory is used to support and explain the main insights obtained.

Overall, it seems that a general recommendation in negotiation procedures is that, regardless the number of players, in the majority of the markets an agreement can be reached when negotiations between producers and retailers is the most observed outcome. It seems that, in real situations, bilateral negotiations between producers and retailers may yield agreement in the distribution of a bundle, although it does not ensure an egalitarian distribution of profits.

Admittedly, our experimental environment had limitations. On the one hand, a limitation of our study is that it was carried out only with students with certain training, and culture and social conditions, but it has not been carried out with students with another type of training, or culture and social conditions. However, the results obtained reveal that some game theory concepts appeared and others did not. In addition, the results show that the structure of the problem is important, which can make the game itself less relevant because information (maybe very relevant for analyzing the situation) is lost, which can justify that some concepts strongly associated with the definition of the game and less with the structure of the problem did not appear, even if they are very intuitive. On the other hand, the fact that some concepts of cooperative game theory did not appear, even though they are very simple and intuitive, may have been due to the students' own training, and culture and social conditions. In this sense, it would be interesting to replicate this experiment in other environments to see the extent to which significant differences appear. Experiments that have been carried out in different cultural, social and training groups have already shown these differences (see, for instance, Henrich [21]).

**Author Contributions:** Conceptualization, J.S.-S.; methodology, N.G.; software, J.S.-S.; validation, N.G., C.G.-H. and S.S J.; formal analysis, J.S.-S. and C.G.-H.; investiga-tion, J.S.-S.; resources, J.S.-S. and C.G.-H.; data curation, J.S.-S.and C.G.-H.; writing—original draft preparation, C.G.-H. and N.G.; writing—review and editing, C.G.-H.; visualization, N.G.; supervision, J.S.-S.; project administration, J.S.-S., C.G.-H. and N.G.; funding acquisition, J.S.-S. and N.G. All authors have read and agreed to the published version of the manuscript.

**Funding:** Carlos Gutiérrez-Hita and Joaquín Sánchez-Soriano acknowledge financial support from the Spanish Government through grant PGC2018-097965-B-I00 funded by MCIN/AEI/10.13039/501100011033 and by ERDF "A way of making Europe"/EU, and from Generalitat Valenciana under project PROM-ETEO/2021/063. Nikolaos Georgantzis acknowledges support by the Spanish Ministerio de Ciencia,

Innovación y Universidades (RTI2018-096927-B-100), Generalitat Valenciana (AICO/2018/036), Universitat Jaume I (UJI-B2018-76/77), Ministerio de Educación Cultura y Deporte (FPU14/02966) and the Agence Nationale de la Recherche (grant ANR-18-CE26-0018). The usual disclaimers apply.

**Institutional Review Board Statement:** Not applicable.

**Informed Consent Statement:** Not applicable.

**Data Availability Statement:** Not applicable.

**Conflicts of Interest:** The authors declare no conflict of interest.

## Appendix A. Instructions for Students (These Instructions are a Translation of the Original Ones in Spanish)

This is a decision-making market experiment. Several groups of students will be formed. Each group consists of different numbers of students. You will participate by assuming the role of a producer (*P*) or a retailer (*R*) in the corresponding market you belong to. If you play the role of a producer, (i) you will have limited capacity in production ($p_i$) of a fictitious good which is non-divisible and perfectly substitutive with the other units produced by the rest of producers in your market, and (ii) you will sell the good to retailers. If you play the role of a retailer, you will have limited demand from consumers ($q_j$) of those perfectly substitutable goods. The trading of a unit of good between a given producer–retailer pair will produce a profit $b_{ij}$ (net of all possible costs involved). If you play the role of a producer, you will be free to produce and sell as many units as you wish up to your maximum capacity of production. If you play the role of a retailer, then you can buy from producers as many units as you wish up to your maximum capacity of sales, given by your consumers demand. A trading plan is a producer–retailer couple $PiRj = x_{ij}$, where $x_{ij} \geq 0$ is the amount of goods traded between producer *i* and retailer *j*. For instance, let us consider a situation with two producers and three retailers.

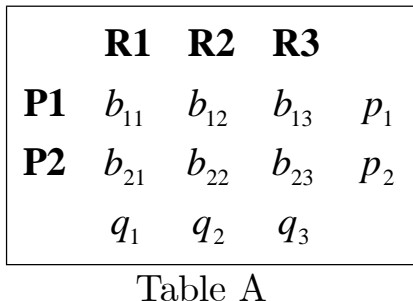

|  | R1 | R2 | R3 |  |
|----|----|----|----|----|
| **P1** | $b_{11}$ | $b_{12}$ | $b_{13}$ | $p_1$ |
| **P2** | $b_{21}$ | $b_{22}$ | $b_{23}$ | $p_2$ |
|  | $q_1$ | $q_2$ | $q_3$ |  |

Table A

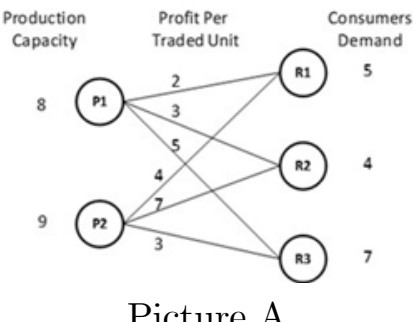

Picture A

Table A resumes your market: number of producers (2), number of retailers (3), maximum production capacity ($p_i$), maximum sales capacity ($q_j$), and the reward for each unit of good traded ($b_{ij}$). Picture A on the right hand side illustrates the transportation problem. For instance, Producer *P1* has a production capacity $p_1$ of 8 units and Retailer *R2* has a sales capacity $q_2$ of 4 units (given by the consumers' demand to that retailer). When *P1* sells one unit of good to *R2* a net profit $b_{12} = 3$ of monetary units (m.u. hereinafter) is obtained. Hence, suppose the producer–retailer pair $P1R2 = 2$, then a net profit of $2 \times 3 = 6$ m.u. is obtained, and so on. . . Your goal, no matter you are either a producer or a retailer is to obtain as much profit as possible.

A *feasible solution* for this problem is a set of trades among the producers and retailers respecting the corresponding constraints. For example, the trade $P1R3 = 6$, $P2R1 = 3$, $P2R2 = 4$ and $P2R3 = 3$ is a non feasible solution: for producer *P2*, $3 + 4 + 3 = 10 > 9$, which exceeds her maximum production capacity. However, trading $P1R3 = 6$, $P2R1 = 3$, $P2R2 = 2$ and $P2R3 = 3$ is a feasible solution. Total net profit obtained with this feasible solution is

$$6X5 + 3X4 + 2X7 + 3X3 = 65 \text{ m.u.}$$

An *optimal solution* is a set of trades among the producers and retailers while respecting the production and consumers' demand constraints, which maximizes the total net profit. In our example, an optimal solution was obtained by trading $P1R3 = 7$, $P2R1 = 5$ and $P2R2 = 4$. Total net profit obtained with this feasible solution is

$$7X5 + 5X4 + 4X7 = 83 \text{ m.u.}$$

One unit of the production capacity of $P1$ is not used. Note that, in general, there could be more than one optimal solution.

The experiment will run for two months. Each group is committed to reaching an agreement on how to distribute the total net profit generated among all the players involved in the problem. You have the freedom to act, negotiate and cooperate among you. If you have some technical or mathematical doubts on the problem, you can ask your lecturer. At the end of the experiment, you must deliver a written report explaining the collective and/or individual procedures followed to achieve the final result, the final trading if any and the final profit obtained by each player.

Your payoff is up to 2 points (out of 10) of the final grade. It is divided into two parts:

1. Written report (75%): The mark of this part will depend on the quality of the analysis of the problem and on the quality of the procedure followed.
2. Your share of total net profits (25%): the mark of this part will be calculated proportionally to the share of the total profit distributed among the members of the group taking into account their contributions to the whole problem in the corresponding transportation situation.

$$Final\ Mark = Mark(1) + Mark(2).$$

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
