# Peer review of "Learning and Applying Cooperative Solutions: A Classroom Experiment on Transportation Games"

_axioms, doi:10.3390/axioms11080397_

Round 1

Reviewer 1 Report

Dear authors:

your study is interesting and relevant. I liked only your conclusion. The other sections of the manuscript need to be revised.
The manuscript needs improvement in terms of structuring (all sections). I give the example of the "Introduction" where it should be comprehensible and sequential:

- background of the field
- general problem this study
- main objective of study
The summary is also weak. It lacks on sentence to put the results into a more general context to provide a broader perspective.
I also give the example of Subsection 2.2.
This subsection is introduced by this sentence: "They will be useful to characterize and analyse our results.” Well, that's not enough, because is not comprehensible to a scientist in any discipline.
The legends of the figures and tables are very confused (for example "Figure 1: Table 2. Type of variable."). I suggest that you make the necessary corrections. In the case of this figure, it is also necessary to separate the categories of variables.

Rew

Author Response

RESPONSE TO REVIEWER # 1

We would like to thank the reviewer for his/her careful reading of our manuscript and for his/her insightful comments and suggestions, which have been of many help to improve the quality of the work. In writing the new version of the manuscript, we have put a lot of effort and care into trying to fulfill all the points raised by the reviewer. Next, we give a detailed answer to each of the points raised by the reviewer.

Comments by the reviewer

Reviewer comment 1. Your study is interesting and relevant. I liked only your conclusion. The other sections of the manuscript need to be revised. The manuscript needs improvement in terms of structuring (all sections). I give the example of the "Introduction" where it should be comprehensible and sequential:

- background of the field
- general problem this study
- main objective of study

Answer: We agree with the referee that the Introduction can be restructured. In the new version of the paper, we follow the sequence you propose: background and literature, the general problem and the objective of the paper (in the same paragraph).

Reviewer comment 2. The summary is also weak. It lacks on sentence to put the results into a more general context to provide a broader perspective.

Answer: Many thanks for this remark. In the new version of the manuscript, we include at the end of the Abstract a general statement based in our findings: “Our findings suggest that bargaining procedures often do not imply explicit coalitions. Moreover, efficiency can be achieved even in the absence of bargaining processes.”

Reviewer comment 3. I also give the example of Subsection 2.2. This subsection is introduced by this sentence: "They will be useful to characterize and analyse our results.” Well, that's not enough, because is not comprehensible to a scientist in any discipline.

Answer: In the new version, we have modified the beginning of the Subsection 2.2 to better explain the main purpose of this section: “Cooperative game theory models situations of conflict of interests in which agents can benefit by collaborating, and binding agreements are possible. In this subsection, we present some basic cooperative game theory concepts which will be used throughout the paper.”

Reviewer comment 4. The legends of the figures and tables are very confused (for example "Figure 1: Table 2. Type of variable."). I suggest that you make the necessary corrections. In the case of this figure, it is also necessary to separate the categories of variables.

Answer: Thanks for this helpful comment. In the new version of the manuscript, we have corrected those legends of Figures and Tables that appeared confused. In particular, we have improved the Figure 1(Table 2) in the way you suggested. The categories of variables are separated by “/” for each variable.

Reviewer 2 Report

1.‎ The article presents a detailed process, thus making it of great interest for the subject.‎ ‎

2.‎ The article is split into logical sections, that makes it easy understand and follow the described ‎process

‎3.‎ It describes key concepts, and useful examples, to better grasp the purpose of the article. ‎

‎4.‎ It properly describes the experiment that the authors set up with the students, and it provided ‎useful definitions, followed by the proper description of the participants and the procedures. ‎

‎5.‎ Data collection is described correctly. ‎

‎6.‎ The results section is divided into several subsections, that aid the understanding of the paper. ‎ ‎7.‎ The formulas are useful and precise.‎ ‎

8.‎ The references are properly described. ‎

‎9.‎ Of particular importance is the Appendix section, dedicated to the students, to help them grasp ‎the importance and the use of the experiment. 

Author Response

RESPONSE TO REVIEWER #2

We would like to thank the reviewer for his/her careful reading of our manuscript and for his/her insightful comments and suggestions. I agree with the referee that English language and style may improve, so we have spell checked the whole manuscript and then had it proofread by a native English speaker.

Reviewer 3 Report

1- The English language is an issue here. For example, the sentence "Within this strand of the literature, in Hong (2019), students where commited to play a classroom trading game where each group is assigned a country. " The first 'where' should be replaced by 'were'. Another example is in " In Swope et al (2014) agents were asked to agree on a credibly bargaining position in laboratory bargaining experiment. " A comma should be after (2014). A comma should be added also after "together with the data collected".

2- We do not put our own findings in the introduction (in the paragraph beginning with ''In our experiment,''. So, these findings should be ommitted from the introduction. 

3- The authors should explain the categories found in Table 2. They could do that after the table. 

4- The authors should define the importand concepts at the beginning, or in the introduction or the literature review. I could not find the definition of  lexicographic or coalition rationality for example. It is not enough to say 'related to ....

5- It is not clear why the authors name  section 4.3 "Results and discussion of the students' behavior". The whole section is results and not discussion. This section should be named 'students' behavior' and the authors should define what they mean by behavior. 

6- The conclusions section appears as a discussion section. The authors should turn this section into a discussion sention. Doing that, they should enrich this section with explanations of the results and related previous studies. 

7- The authors should add a section named "conclusions, limitations and recommendations. 

Author Response

RESPONSE TO REVIEWER #3

We would like to thank the reviewer for his/her careful reading of our manuscript and for his/her insightful comments and suggestions, which have been of many help to improve the quality of the work. In writing the new version of the manuscript, we have put a lot of effort and care into trying to fulfill all the points raised by the reviewer. Next, we give a detailed answer to each of the points raised by the reviewer.

Comments and Suggestions for Authors

Reviewer comment 1.  The English language is an issue here. For example, the sentence "Within this strand of the literature, in Hong (2019), students where committed to play a classroom trading game where each group is assigned a country." The first 'where' should be replaced by 'were'. Another example is in "In Swope et al (2014) agents were asked to agree on a credibly bargaining position in laboratory bargaining experiment." A comma should be after (2014). A comma should be added also after "together with the data collected".

Answer: We agree with the reviewer that a complete revision of the English style is advised. In the new version of the manuscript, we have corrected all the points raised by you. Moreover, we have conducted a professional revision of the English style.

Reviewer comment 2. We do not put our own findings in the introduction (in the paragraph beginning with ''In our experiment,''). So, these findings should be omitted from the introduction. 

Answer: Thank you for this comment. We have moved the main findings in this paragraph to the Section conclusions. However, the first part of the paragraph remains in the Introduction as it highlights the general setting of the experiment, not the main findings. Moreover, the structure of the Introduction has been modified in order to fulfill a comment raised by referee #1. Finally, the Section Conclusions have been modified because of another comment pointed out by reviewer #1 (in the new version, there is a new section before Conclusions where we discuss the main findings of the paper).

Reviewer comment 3. - The authors should explain the categories found in Table 2. They could do that after the table. 

Answer: Thank you for this suggestion. In the new version of the manuscript, following your suggestion, we have explained the categories (i.e., we explain each variable in order to better understand the categories they can take). You can see that there is a new paragraph and that the paragraph beginning “First, we focus…” has been modified in some parts.

Reviewer comment 4. The authors should define the important concepts at the beginning, or in the introduction or the literature review. I could not find the definition of lexicographic or coalition rationality for example. It is not enough to say 'related to ....

Answer: We agree in that point with the referee. In the new version of the manuscript, following your suggestion, we have included the definition of lexicographic orders in the subsection Basic concepts (point 6), and coalition rationality (after Table 2).

Reviewer comment 5. It is not clear why the authors name section 4.3 "Results and discussion of the student behavior". The whole section is results and not discussion. This section should be named 'students' behavior' and the authors should define what they mean by behavior.

Answer: Many thanks for this remark. In the new version of the manuscript, we rename the section simply as Behavior. Moreover, we have included a definition of behavior in our experimental context: “In our context, behavior describes the type of strategies that a student (or a coalition of students) follow when they define a trading plan in order to capture as much as profits they can achieve.”

Reviewer comment 6. The conclusions section appears as a discussion section. The authors should turn this section into a discussion section. Doing that, they should enrich this section with explanations of the results and related previous studies.

Answer: Thank you for this suggestion. In the new version of the manuscript, we rename the section as Discussion of results. In this section, we include a discussion of our results related with other previous studies. Accordingly, the section Conclusion has been rewritten.

Reviewer comment 7. The authors should add a section named "conclusions, limitations and recommendations.

Answer: In the new version of the manuscript, we have included a new section labelled “Conclusions, recommendations and limitations”.

Round 2

Reviewer 3 Report

Thanks for the revision.